# Augmentation Method for High Intra-Class Variation Data in Apple Detection

**DOI:** 10.3390/s22176325

**Published:** 2022-08-23

**Authors:** Huibin Li, Wei Guo, Guowen Lu, Yun Shi

**Affiliations:** 1Institute of Agricultural Resources and Regional Planning, Chinese Academy of Agricultural Sciences, Beijing 100081, China; 2Graduate School of Agricultural and Life Sciences, The University of Tokyo, Tokyo 188-0002, Japan; 3College of Urban and Environmental Sciences, Central China Normal University, Wuhan 430079, China

**Keywords:** deep learning, modern orchards, visual detection, fruit picking, lightweight detection models, augmentation method

## Abstract

Deep learning is widely used in modern orchard production for various inspection missions, which helps improve the efficiency of orchard operations. In the mission of visual detection during fruit picking, most current lightweight detection models are not yet effective enough to detect multi-type occlusion targets, severely affecting automated fruit-picking efficiency. This study addresses this problem by proposing the pioneering design of a multi-type occlusion apple dataset and an augmentation method of data balance. We divided apple occlusion into eight types and used the proposed method to balance the number of annotation boxes for multi-type occlusion apple targets. Finally, a validation experiment was carried out using five popular lightweight object detection models: yolox-s, yolov5-s, yolov4-s, yolov3-tiny, and efficidentdet-d0. The results show that, using the proposed augmentation method, the average detection precision of the five popular lightweight object detection models improved significantly. Specifically, the precision increased from 0.894 to 0.974, recall increased from 0.845 to 0.972, and mAP0.5 increased from 0.982 to 0.919 for yolox-s. This implies that the proposed augmentation method shows great potential for different fruit detection missions in future orchard applications.

## 1. Introduction

Deep learning has evolved rapidly in recent years. A proper deep learning algorithm requires a suitable dataset or data augmentation method to demonstrate its superior performance. Therefore, finding pre-processing methods to enhance data quality is a critical step in research since it is more effective to increase the quantity, diversity, and quality of a dataset than to increase the complexity and depth of a model [1], not only to increase the generalization capability of the algorithm but also to make research more convincing.

Research related to data quality improvement is essential for fruit-picking detection in agriculture. It is well-known that the significant demand for mechanized automatic fruit harvesting in the agricultural sector offers significant opportunities for developing agricultural picking robots. Automatic picking robots have received a great deal of attention from researchers in recent decades, and a variety of robots have been developed domestically and internationally to harvest fruits and vegetables, such as apple-picking robots [2]. In addition, vision-based control technology for picking robots has seen rapid development in recent years. Although there has been a great deal of research in the development of vision-control techniques for robotic picking, the low success rate of fruit recognition and inefficient hand–eye coordination still limits picking robot performance [3]. Occlusion is considered to be one of the challenges of robotic vision picking technology since it seriously affects the recognition and localization accuracy of picking robots [4], mainly in the case of leaf and branch occlusion and when fruit overlaps, resulting in a long time requirements for target fruit recognition, low recognition accuracy, and more difficult recognition at night [5]. Recent advances in deep learning have greatly improved the perception of occlusion fruit because relevant detection algorithms are sensitive to changes in object appearance and environmental conditions [6]. Research related to improving the quality of training data can further facilitate the optimal performance of deep learning detection algorithms.

In researching a detection model for apple-picking robots, the model needs to determine multi-type occlusion generated by branches, leaves, and fruits to improve the picking ability of the robot to select fruit. In various studies, apple-picking robots only detected apples as one class [4,7,8,9,10,11,12,13,14,15,16,17,18,19,20,21,22,23,24], ignoring apple occlusion during picking, which can potentially damage the end-effector and target apples [25,26]. Therefore, fine detection of multi-type occlusion fruits is required prior to picking, which is a much more complicated method and requires detection based on fine features of the apple position. In studies performed to detect multi-type occlusion fruit, a significant variation in sample size between classes in most datasets is common [27,28]. We also found high intra-class variation when the collected raw images were annotated and counted by multi-type occlusions. Moreover, classification bias or algorithm performance degradation will occur in the process of model detection due to the imbalance of annotation numbers in each type of occlusion, which affects the effectiveness and accuracy of the trained model. For example, the algorithm overfits classes with too many samples and underfits classes with very few samples. It is necessary to have methods that can solve the problem from a data perspective, including oversampling and undersampling, to reduce the effects of the uneven number of training samples of each type on the training of the model [29]. The method presented in [30] derived the result that both oversampling and undersampling increased model recall by using multiple resampling algorithms, with oversampling improving the detection performance of all models and being one of the common methods used in deep learning [31,32]. Oversampling methods generally replicate samples from a small number of classes to fill the number. To reduce the overfitting problem of the model, the method in [33] used a combination of inter-class weight assignment, central loss, and adaptive synthetic sampling approaches [34] to achieve a balance in the number of samples for each type of pest and disease, which eventually further reduced model loss during model training, and the detection accuracy of each type of pest and disease was greatly improved. Undersampling methods involve randomly removing a large number of available sample classes to equal the number of classes [35]. While these methods can be helpful in some cases, randomly discarding data reduces the total amount of learning data required by the model and may prevent the model from performing optimally. In contrast, some methods modify the way the model learns during training, focusing less on the majority groups and more on the minority groups [36].

In this study, we first build the multi-type occlusion apple (MTOA) dataset, then propose a balance augmentation method. The method is based on the differences between the number of annotation boxes of each apple occlusion class and the pattern of occlusion existence for manual data synthesis. Moreover, this method achieved a balanced number of annotation boxes of each apple occlusion class in different regions under different illuminations. Five lightweight models (i.e., yolox-s, yolov5-s, yolov4-s, yolov3-tiny, and efficientdet-d0) were used as basic models to verify the effectiveness of the proposed method. Each model was trained with basic training datasets of MTOA and balanced MTOA datasets. Finally, the performance metrics were compared. The results show that the proposed method could solve the imbalance problem in the number of annotation boxes in the training part of the MTOA dataset. In addition, it could avoid the annotation noise and overfitting phenomena.

The highlights of this study are as follows:We created the MTOA, the first dataset considering multi-type occlusion of apple fruits, and made it available for free under the MIT license.We proposed a balance augmentation method to achieve a balanced number of annotation boxes of each apple occlusion class in different regions under different illuminations and solved the problem of severe differences in the number of samples between classes.We validated the effectiveness of the proposed algorithm using five popular lightweight object detection models.

## 2. Materials and Methods

### 2.1. Making the MTOA Dataset

The raw images in the apple orchards were obtained by self-collection and web-collection, respectively, and an example of the collected images for each region is shown in Figure 1. The raw images include images of Yanfu-3 apples from Zhaoyuan, Shandong, China (SD_ZY_IMG), Yanfu-8 apples from Qixia, Shandong, China (SD_QX_IMG), and Jonagold apples from Prosser, Washington, USA (WT_PSR_IMG) [25]. The specific information about the collected images is listed in Table 1.

There is variability in the way orchards are grown and data collected between the three regions. The Zhaoyuan orchard is a modern spindle planting structure with apple trees spaced in rows approximately 3.5 m apart, with 1.5 m between plants, and with a height of 3.5 m. Multi-angle photography was mainly performed by handheld cameras at a distance of about 0.5–1.0 m. The images were taken in the morning, at midday, and in the evening, with clear weather during the day and artificial lighting at night. On the other hand, the Qixia apple orchard is a traditional orchard with a happy canopy, with rows about 4 m apart, plants approximately 5 m apart, and trees with a height of about 3 m. It was mainly photographed by handheld cameras at a distance of 0.3–0.8 m, at midday with clear weather. Meanwhile, the Prosser apple orchard in Washington State has a tree-wall structure. The data on Jonagold apples were collected by mounting the camera, which was approximately 1.7 m above the ground, on a mobile platform and keeping the distance from the camera to the tree wall at about 0.5 m at midday with clear weather.

Since there are no publicly available apple datasets, we manually annotate all images. Annotation classes consisted of eight types of occlusions, including no occlusions (N), leaf occlusions (L), fruit occlusions (F), branch occlusions (B), leaf and fruit occlusions (LF), leaf and branch occlusions (BL), branch and fruit occlusions (BF), and leaf, branch, and fruit occlusions (BLF). Each annotation class is shown in Figure 2. The MTOA dataset was constructed after all annotations. We counted all annotation boxes, and the result showed that the class with the largest number of annotation boxes was N (22986), accounting for 28.2% of the total, and the class with the smallest number of annotation boxes was BLF (1374), accounting for 1.7% of the total, which is a difference of approximately 16 times. In BF, F, and LF, the proportion of data in each occlusion class did not exceed 5% of the total number of annotated boxes, which shows the significant variability between all occlusion apple classes in the MTOA dataset.

### 2.2. Data Balance Algorithms

#### 2.2.1. Algorithm Validation Process

The validation flow of the data balance augmentation algorithm proposed in this study is shown in Figure 3. The diagram contains the following steps.
Splitting the MTOA dataset into a basic test and training dataset with a ratio of 3:7 and then training five lightweight models with the basic training dataset. The training of five lightweight models was carried out to form five corresponding basic models.Balancing the basic training dataset using the proposed method to form the balanced MTOA dataset.Training five lightweight models with the balanced MTOA dataset. Five lightweight models were trained to form five corresponding balanced models.Using basic test dataset to perform metrics on five basic and balanced models and analyze reasons for changes in model performance.

#### 2.2.2. MTOA Dataset Statistical Analysis by Illumination

The lighting methods in the apple orchards were natural lighting during the day and artificial lighting at night. Different lighting methods produce different light illuminations (i.e., RGB vectors of different lighting colors) [37]. As illumination changes, the same scene shows different color representations, for example, backlit and artificially lit images in which apple targets tend to be too bright or have severe color distortion. In this study, to effectively classify images under different illuminations [38], a MobileNetV3-based classification model was used to classify raw images from each region into high and low illuminations, with low-illumination images mainly containing images under backlight, evening, or nighttime artificial lighting conditions, and high-illumination images containing images under sufficient daytime light conditions. The classification of raw images by illumination is shown in Appendix A.

The MTOA dataset was analyzed and counted by illumination to clarify the number of each type of annotation box in each region with high and low illuminations in the MTOA dataset. First, raw images in the MTOA dataset were classified using the illumination binary classification model of MobileNetV3 to form six sub-datasets, including the ZhaoYuan high-illumination dataset (ZY_H), ZhaoYuan low-illumination dataset (ZY_L), Qixia high-illumination dataset (QX_H), Qixia low-illumination dataset (QX_L), Prosser high-illumination dataset (PSR_H), and Prosser low-illumination dataset (PSR_L). Six sub-datasets were counted by illumination for each occlusion type, and the results are shown in Table 2.

The statistics showed that QX_L had a low number of annotation boxes and PSR_L had zero annotation boxes because most of the raw images in QX_L and all raw images in PSR_L were collected in the morning with even lighting, respectively. Due to the small number of annotation boxes in QX_L and PSR_L, data balance augmentation of these two sub-datasets was ignored. From the other four sub-datasets, the number of annotation boxes varied greatly between apple occlusion classes, with the largest class being B for ZY_H, the smallest class being BF for PSR_H, and the difference between them being approximately 195 times. In summary, the number of annotation boxes and data in sub-datasets by illumination varied greatly between regions, resulting in difficulty making the training data from different regions and different illuminations play an equal role in calculating the training loss of the model.

#### 2.2.3. Rules for Building the Component Pool

The component pool is a collection of elements required for synthesizing each occlusion apple. It consists of five elements in high and low illuminations: base image elements, fruit elements, branch elements, leaf elements, and composite elements.
Rules for making base image elementsIn this study, images with no fruit in high illumination and matching the orchard background were selected as high-illumination base images. Images with no fruit in low illumination and matching the orchard background were selected as low-illumination base images. However, since there were fewer images in this study, some low-illumination images were blurred and supplemented as base images. These two types of base images were randomly scaled to 640 × 480 and 1280 × 720 to maintain consistency with the image size in basic training dataset. Finally, 1000 high-illumination base images and 1000 low-illumination base images were selected. Figure 4 shows various base images under high and low illuminations.Rules for the selection of occlusion elementsIn this study, fruit, branch, leaf, and composite occlusion elements were segmented from images from the basic training dataset. The fruit occlusion elements were mainly single intact fruits; the branch elements were divided into multiple branches and single branches; the leaf elements were divided into a single leaf and multiple leaves; and the composite elements were mainly the combination of branches, leaves, and fruits. Five hundred elements of each of the five classes were segmented to ensure the diversity of selected results. After that, fruit, branch, leaf, and composite occlusion elements under high illumination were grouped into one category. Meanwhile, the fruit, branch, leaf, and composite occlusion elements under low illumination were grouped into another category. Figure 5 shows the occlusion elements at high and low illuminations.

#### 2.2.4. Data Synthesis Methods for Each Apple Occlusion Class

The main idea in synthesizing each occlusion apple class was to paste the corresponding occlusion elements into the N area of the raw image, depending on the number of occlusion apple classes to be synthesized and the illumination requirements. This was because the occlusion classes could be combined from N and occlusion elements after observing raw images. All occlusion elements entered from the boundary region of N and extended randomly to arbitrary locations. This study used this prior knowledge to complete the synthesis of each occlusion apple class. There was also variability in the synthesis of each occlusion class. For B and L, both could be formed by randomly pasting leaf and branch elements into N according to the edge entry rule. However, for F, they could not be synthesized according to this method because if N was shaded by more than 50%, the fruit occlusion element easily became an N and the shaded fruit became F (i.e., both occlusion apple classes would appear simultaneously). The BL, BF, LF, and BLF were composite occlusion classes and could be formed by either synthesizing composite occlusion elements or attaching the existing composite occlusion elements to N.
N, B, and L class synthesis methodsN synthesis required the cyclic extraction of the required number of N from the basic training dataset for the corresponding region and illumination. B and L were formed by directly attaching branch and leaf occlusion elements to the surface of N randomly according to the edge entry rule. Figure 6 shows the synthesis of B and L.
First, the numbers of B and L to be synthesized were calculated. Then, the N class was selected from the images of the basic training dataset for the corresponding region and illumination. The selected N was divided into six equal parts according to their width and height to form a grid with a size of 6 × 6. Since all occlusion elements were entered by the edges of N, the boundary of the grid was set in this study as an edge entry area, within which the starting points of branch and leaf needed to be selected.The branch and leaf elements were randomly extracted from the component pool and cropped at a scale of 0.5–1.0 to form a new occlusion element.The edge entry area contained 24 location points from which the starting point of the occlusion element was randomly selected. The endpoint of the occlusion element could not be in the same row or column as the random starting point because the area formed by the starting and endpoints of the occlusion element in the same row and column was a line, which cannot provide a rectangular area of the same size for the occlusion element. To highlight the occlusion elements, this study set the distance between the starting and endpoints of the occlusion element to be greater than three grid lengths to ensure that any area adjacent to the random starting point that was less than three grid lengths could not be used as the endpoint of the selection area. Other areas could be used as the endpoint of the selection area. Then, we randomly selected the endpoint of the occlusion elements from the endpoint of the selection area.After determining the random starting points and random endpoints of the occlusion element, the dimensions of the new occlusion element were transformed by scaling or cropping the rectangular area between the random start and endpoints. Then, the changed occlusion element was pasted in the random starting and ending areas to form B or L, and finally, the synthetic B or L image was saved.Class F synthesis methodWhen synthesizing class F, it was impossible to directly attach the fruit occlusion element to the N surface. This was because it was easy to identify the fruit occlusion element as N during the model training process if the N area was overshadowed. The synthesis process of F, shown in Figure 7, was accomplished by limiting the common area of the fruit occlusion element and N.
First, N was selected from the images of the basic training dataset for the corresponding region and illumination, which depended on the number of F to be synthesized. Then, we obtained the fruit occlusion element from the component pool according to illumination demand and adjusted its size to the same size as the selected N. Subsequently, we divided N into a 6 × 6 grid and created a 14 × 14 grid area with the center point of N as the origin. The 14 × 14 grid area was divided into four quadrants, with the upper left corner as the first quadrant.The location of the upper left corner of the fruit occlusion element was randomly selected in the first quadrant. To prevent the fruit occlusion element from completely obscuring N, the centroid of the occlusion element and origin were not allowed to overlap. This method limited the area of overlap between the fruit occlusion element and N to no more than 34%, because the fruit occlusion element would become N if it exceeded 34%, leading to confusing annotations.After determining the starting position of the upper left corner for the fruit occlusion element, the upper left corner of the fruit occlusion element and starting position were overlapped to complete the operation of pasting the fruit occlusion element on a 14 × 14 grid. Finally, area N on a 14 × 14 grid was intercepted, and the result of the interception was the synthesized F image.Fused occlusion-type compositing methodsThe fused occlusion apples mainly consisted of BL, BF, LF, and BLF. All four could be synthesized on the basis of B, L, and F by the attachment of a second or third occlusion element to form the final fused occlusion apple class. The results of BL, BF, LF, and BLF are shown in Figure 8, and the specific synthesis process is described in Method 1 of the Appendix A.

#### 2.2.5. Making the Augmented MTOA Dataset

After synthesizing all occlusion apple images, the next step was to attach these images to base images and automatically label them to form the augmented MTOA dataset, as shown in Algorithm 1. Finally, the augmented MTOA and basic training datasets were combined to form the balanced MTOA dataset.
**Algorith****m 1:** Making the Augmented MTOA Dataset**Input:**a. Synthetic occlusion apple images: D=Arealid{dimgsji}  b. Base images under high and low illumination: B=Arealid{b_imgsk}**Output****:** Balanced MTOA dataset: DB=Arealid{db_imgs,db_labels}
      for area←Zhaoyuan to id  do               for  lgt←high to l  do         b_img_id=Random(0,k−1)                            b_img=Arealid{b_imgsk}(area,lgt,b_img_id)                          for  cls←N to i do         for  img_id←1 to j do                   shelter_img=Arealid{d_imgsji}(area,lgt,cls,b_img_id)                  if bimg.remainingspace>shelterimg.shape  then                      start_pos=random(x,y)  in b_img.remaining_space                    end_pos=(x+shelter_img.shape.x,y+shelter_img.shape.y)                      copy shelterimgto bimg.remainingspace with strartpos                        update b_img.remaining_space                                             labels+=(cls,strart_pos,end_pos)                                  else    save new bg_img                       DB(area,lgt)=(bg_img,labels)                               clear labels*id**∈**(Zhaoyuan,Qixia,Procieer), l**∈**(low,high), i**∈**(N,L,F,B,LF,BL,BF,BLF), j is num of class i with l light, k is img num with l light*.

### 2.3. Equipment and Model Training

#### 2.3.1. Training Equipment

All experiments were conducted on a deep learning server with Intel Xeon(R) Gold 6226R v4@2.90 Hz × 64 CPU, 251.6 GB RAM, 1.9 TB SSD, 2 × 16 GB NVIDIA Tesla V100 and software: Ubuntu 20.04 OS, PyTorch 1.10 (America, Facebook), NVIDIA driver 495.46, CUDA 11.5 and cuDNN 8.2.4 (America, Nvidia). 

#### 2.3.2. Model Selection and Training

In this study, from the perspective of the practical use of agricultural robots with embedded computing, the primary consideration is the ability of the algorithm to detect fruits quickly and accurately in real time [39]. Therefore, five lightweight models that could be deployed in embedded AI terminals were used to verify the effectiveness of the algorithm. These models were yolox-s, yolov5-s, yolov4-s, yolov3-tiny, and efficientdet-d0, with fps of 73, 73, 164, 556, and 31, respectively, on the experimental host. The specific information for each model is listed in Table 3. The training progress was stopped when the accuracy of each model reached convergence, and the optimal model was saved at the end of training. The model training time was measured in hours, and each model contained two training times; the first value was the training time using the MTOA dataset, and the second value was the training time using the balanced MTOA dataset.

### 2.4. Performance Metrics

In this study, four metrics were used to evaluate the performance of the trained model, including precision (P), recall (R), average precision (AP), and mean average precision (mAP), which were calculated as follows:(1)P=TP/(TP+FP),
(2)R=TP/(TP+FN),
(3)$$AP\left(n\right)=\int_{0}^{1}P_{n}\left(R_{n}\right)dR_{n},$$
(4)$$mAP=0.125*\overset{8}{\underset{n=1}{\sum }}AP\left(n\right),$$
where P is the proportion of the correct prediction boxes detected among all prediction boxes; R is the proportion of the correct prediction boxes among all annotation boxes; AP is the average accuracy value for each occlusion apple class that measures how well the trained model does on each class; mAP is the average AP value for eight occlusion apple classes to measure how well the trained model does on all classes; TP is the number of correctly matched prediction boxes for all annotation boxes; FP is the number of incorrectly predicted boxes; and FN is the number of missed annotation boxes.

## 3. Results

### 3.1. Results after the Balance of Each Occlusion Class

The basic training dataset was balanced to ensure that the numbers of annotation boxes in each occlusion apple class were equal. However, annotation boxes in QX_L and PSR_L were not balanced because of their small number. The final number of annotation boxes in each occlusion apple with each illumination was 10,579, which is the number of B in ZY_H. The total number of annotation boxes was 338,937, and the original number of annotation boxes was 81,631. The results of the synthesis of the images under high and low illuminations are shown in Figure 9, and the numbers of annotation boxes of each occlusion class after synthesis are shown in Table 4.

### 3.2. Training Results before and after Data Balance

In this study, the yolox-s-basic, yolov5-s-basic, yolov4-s-basic, yolov3-tiny-basic, and effidentdet-d0-basic models were obtained after training five lightweight models using basic training dataset from the MTOA dataset, and we obtained yolox-s-bal, yolov5-s-bal, yolov4-s-bal, yolov3-tiny-bal, and efficidentdet-d0-bal after training five lightweight models using the balanced MTOA dataset. Since all models were pre-trained using the balanced MTOA dataset, it was found that models trained without mosaic and mixup augmentation algorithms performed better than those trained with such algorithms, indicating that the balance augmentation method is more effective than mosaic and mixup methods when training models for apple occlusion detection. This is mainly because the feature difference between the occlusion targets is often in a small local area, and the mosaic and mixup augmentation algorithms increase annotation noise. The test dataset in this study consisted of the basic_test and balance_test datasets, both of which were identical. The test dataset contained 1000 images and labels under ZY_H, ZY_L, QX_H, and PSR_H, excluding synthetic images, for a total of 12,666 apple occlusion targets. The test metrics for each model were calculated using Equations (1)–(4), as shown in Table 5, and the following information was obtained from the analysis of the results.
Yolox-s-bal improved significantly over the yolox-s-basic model in terms of precision, recall, and mAp, and it showed some improvement in Ap values between all occlusion classes. Moreover, yolov4-s-bal, yolov3-tiny-bal, and yolox-s-bal had the same performance, indicating that the balance method proposed in this study can improve the performance of the above four models in detecting apple occlusion targets in a comprehensive manner. Compared with the yolov5-s-basic model, the precision and AP_BLF_ values decreased by 0.001 and 0.042, respectively. However, the recall value improved significantly, the mAp value remained balanced, and the AP values of all classes except BLF improved, indicating that the proposed method in this study can maintain an accurate performance of yolov5-s-basic while improving the ability of the model to find a full range of occlusion targets.The highest values of each metric in the detection results were P (0.974) in yolox_s_bal; R (0.972) in yolox-s-bal; mAp (0.960) in yolov5-s-basic and yolov5-s-bal; AP_N_ (0.985) in yolov5-s-bal; and AP_B_ (0.982), AP_L_ (0.975), AP_F_ (0.963), AP_BL_ (0.965), AP_BF_ (0.959), AP_LF_ (0.956), and AP_BLF_ (0.994) in yolox-s-bal, showing that the highest metric values were among the models trained with the balanced MTOA dataset. This indicates that the proposed method can promote a great expression of the detection performance of the model for some categories.The proposed method improved most of the metrics of each model, but the AP_BLF_ metrics decreased after the training of yolov5-s-bal and efficientDet-d0-bal, indicating that the features of the BLF in the balanced MTOA dataset differed from those in the BLF classes of the basic training dataset. This affected the detection ability of the BLF in these two models.

## 4. Discussion

The experimental results show that the AP_BLF_ value decreased for the yolov5-s-bal and efficientDet-d0-bal models and increased for all other models. To analyze the reasons, the original test dataset was divided into four sub-classes of test datasets (ZY_H_TEST, ZY_L_TEST, QX_H_TEST, and PSR_H_TEST). Then, the yolov5-s-bal model was used as a representative of an AP_BLF_ decreasing model because the AP_BLF_ value in yolov5-s-bal declined the most, and yolov4-s-bal was used as a representative of a rising model for all metrics. Finally, these two models were tested separately with four sub-classes of test datasets, as the AP_BLF_ value in yolov4-s increased the most.
First, four sub-classes of test datasets were tested based on the yolov5-s-basic and yolov5-s-bal models. Test results are shown in Table 6, and visualization test results are shown in Appendix A. In the test results for ZY_H_TEST and QX_H_TEST, the AP_BLF_ improved by 0.018 and 0.071, respectively, but the AP_BLF_ decreased by 0.085 and 0.096 in the test results for ZY_L_TEST and PSR_H_TEST, respectively. Meanwhile, in the test results for PSR_H, all metrics decreased for different illuminations except AP_F_, indicating that the yolov5-s-bal model had a loss in the detection performance of the BLF in PSR_H_TEST and ZY_L_TEST, which is also the main reason for the decrease in AP_BLF_ when testing the yolov5-s-bal model. In addition, the yolov5-s-basic model had the greatest variability in all test metrics for PSR_H_TEST. If we want to continue to improve the metrics, we need to provide data that are more similar to source data for model training. However, for PSR_H_TEST balance augmentation, the occlusion elements in ZY_H and ZY_L were used for synthesis, which improved the amount of training data but fell short of the goal of being more similar to the source data, resulting in a decrease in the AP_BLF_ and other metrics.Four sub-classes of test datasets were tested based on the yolov4-s-basic and yolov4-s-bal models. The test results are shown in Table 7, and the visualization test results are shown in Appendix A. The decreases in AP_BLF_ and P for ZY_L_TEST data were 0.036 and 0.001, respectively, with a large decrease in AP_BLF_, which is due to the same reason as the decrease in AP_BLF_ for the yolov5-s-bal model test. Moreover, the detection metrics under all other sub-class test datasets showed substantial improvements, indicating that the combined detection performance of the yolov4-s-bal model improved significantly. Although the detection ability of BLF in ZY_L_TEST data was suppressed, the AP values in other sub-classes increased significantly, which was the main reason for the increase in AP_BLF_ when all test datasets passed the yolov4-s-bal test. The results of the yolov4-s-basic model for PSR_H_TEST, ZY_H_TEST, and PSR_H_TEST were all relatively low, which shows that there is more room to improve the performance of the model. All other metrics improved after training the model with a balanced MTOA dataset, indicating that the proposed method in this study is more effective in improving the performance of the model when there is more room for improvement.

The method proposed in this study is expected to be applied to the fine detection of apple fruit in multiple regions. Compared with most data augmentation algorithms, the proposed method greatly increases the quantity and quality of underlying multi-type occlusion apple data and solves the problem of unbalanced quantities between classes. The detection capability of the lightweight model and the generalization capability of the model were also improved, which shows that our method can be used for advancing the fine-grained identification of occlusion fruits. This method needs to be used to fully use a priori knowledge (consistency of occlusion elements, type of occlusion, and features of external illumination) to generate high-quality data and ensure that synthetic data are consistent with the main features of the original data.

## 5. Limitations and Future Research

Despite the aforementioned achievements [40], there is still room for improvement in the proposed method. Firstly, the process of component pool production consumes a great deal of time because all elements need to be manually selected or extracted, such as selecting base images, manually segmenting fruits, leaves, branches, etc., which increases the cost of data production. Secondly, after the MOTA dataset is balanced, the size of the balanced MOTA dataset will increase, and the training time of the model will increase accordingly when more data are used for model training. In the future, our research direction is to reduce the dataset production time and labor cost by using the existing segmentation model for building a component pool.

## 6. Conclusions

In this study, we addressed the problem that most lightweight models detect multiple types of occlusion targets inefficiently during fruit picking. We proposed the first MTOA dataset and a balance augmentation method. The results show that using the proposed method, the average detection precision of the five popular lightweight object detection models can be significantly improved, demonstrating the effectiveness of the proposed method. However, we still need to pay attention to the selected occlusion types that should be consistent with the actual situation since this will affect the similarity between the synthetic and actual data. The proposed method showed considerable potential for different fruit detection missions in future orchard applications in complex environments.

## Figures and Tables

**Figure 1 sensors-22-06325-f001:**
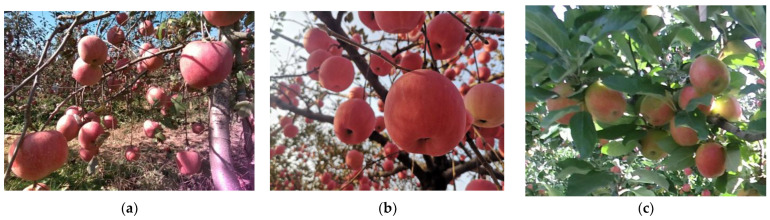
Example of the raw images in the MTOA dataset: (**a**) SD_ZY_IMG; (**b**) SD_QX_IMG; (**c**) WT_PSR_IMG.

**Figure 2 sensors-22-06325-f002:**
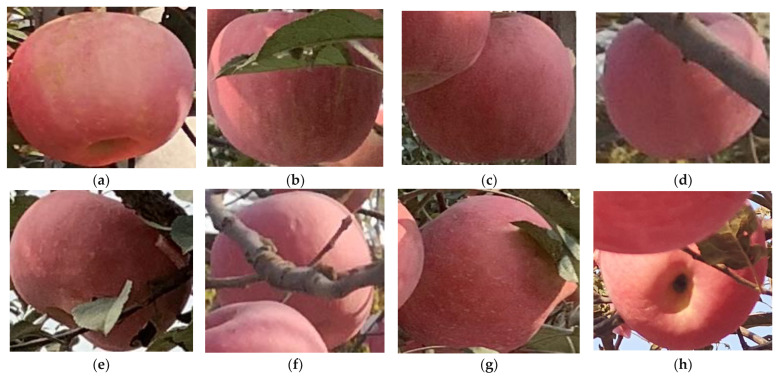
Annotation classes: (**a**) no occlusion; (**b**) leaf occlusion apples; (**c**) fruit occlusion; (**d**) branch occlusion; (**e**) leaf and fruit occlusion; (**f**) leaf and branch occlusion; (**g**) branch and fruit occlusion; and (**h**) leaf, branch, and fruit occlusion.

**Figure 3 sensors-22-06325-f003:**
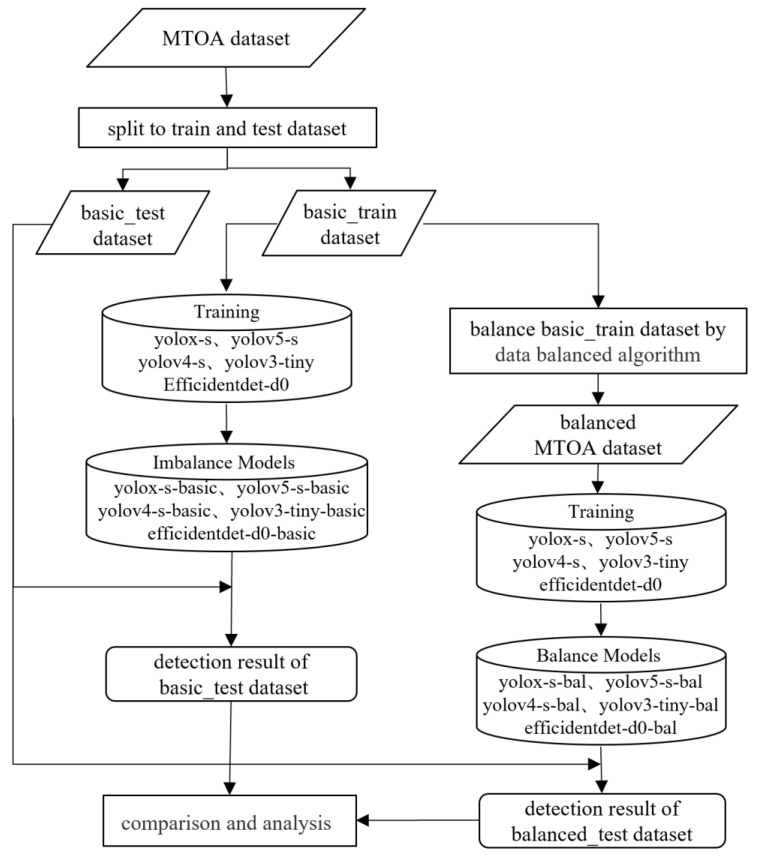
Flow chart for the validation of the proposed method.

**Figure 4 sensors-22-06325-f004:**
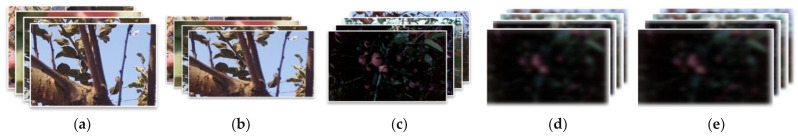
Various base images with high and low illuminations: (**a**) 640 × 480 base image under high illumination; (**b**) 1280 × 720 base image under high illumination; (**c**) 1280 × 720 base image under low illumination; (**d**) 640 × 480 blurred base image under low illumination; (**e**) 1280 × 720 blurred base image under low illumination.

**Figure 5 sensors-22-06325-f005:**
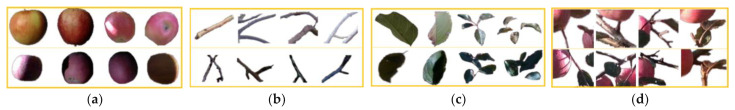
Various occlusion elements under high and low illuminations: (**a**) fruit occlusion elements under high and low illuminations; (**b**) branch occlusion elements under high and low illuminations; (**c**) leaf occlusion elements under high and low illuminations; (**d**) combined leaf, branch, and fruit occlusion elements under high and low illuminations.

**Figure 6 sensors-22-06325-f006:**
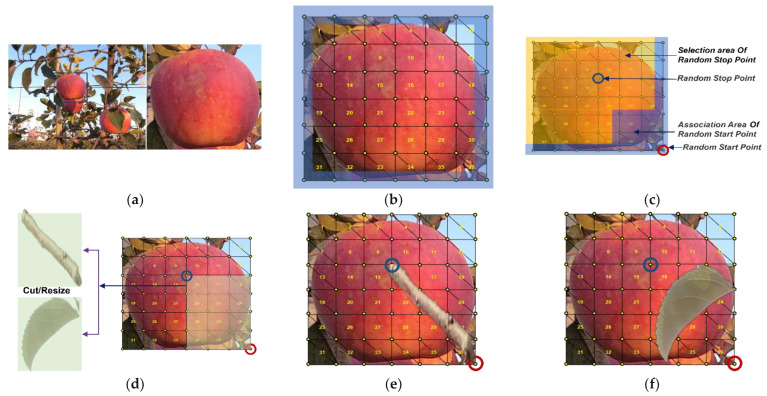
The synthesis process of B and L: (**a**) N is selected from the annotated images. (**b**) N area is gridded, and the blue part is the edge entry area. (**c**) The gridded area is divided into a random element insertion point selection area and a random element endpoint selection area, with the red point being the random starting point and the blue point being the random endpoint. (**d**) Branch or leaf occlusion elements are inserted between the random starting and random endpoints. (**e**) Synthetic B image; (**f**) Synthetic L image.

**Figure 7 sensors-22-06325-f007:**
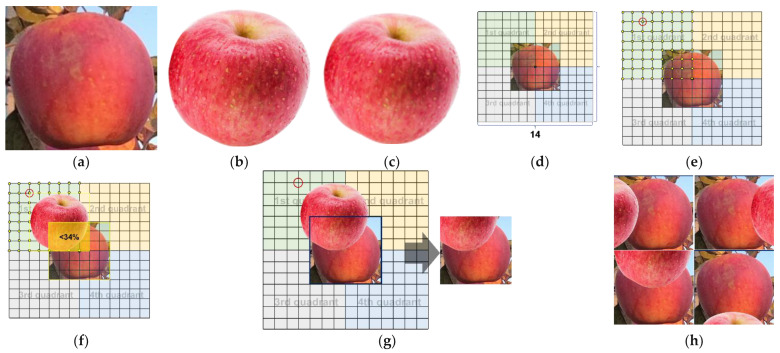
The synthesis process of F: (**a**) N is selected from annotated images. (**b**) Fruit occlusion element is selected from the component pool. (**c**) Fruit occlusion element is resized to N size. (**d**) N is divided into 6 × 6 grid regions, a 14 × 14 grid region is created with the center of N as the origin, and it is divided into four quadrants. (**e**) The randomly selected start point of the fruit occlusion element is based on the first quadrant, which is a red circle. (**f**) The fruit occlusion element is attached to the 14 × 14 grid area based on the random starting point, ensuring that the overlap between the fruit occlusion element and N is less than 34% of N area. (**g**) The original image is cut in according to the size of N to form F. (**h**) The other four F.

**Figure 8 sensors-22-06325-f008:**
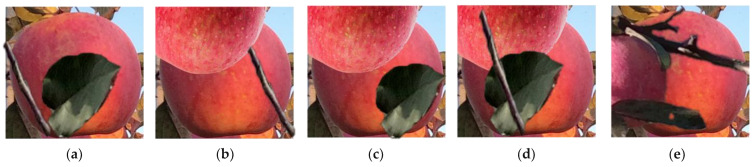
Synthesis results of fused occlusion apple: (**a**) Synthetic BL image; (**b**) Synthetic BF image; (**c**) Synthetic LF image; (**d**) Synthetic BLF image; and (**e**) Synthetic BLF image.

**Figure 9 sensors-22-06325-f009:**
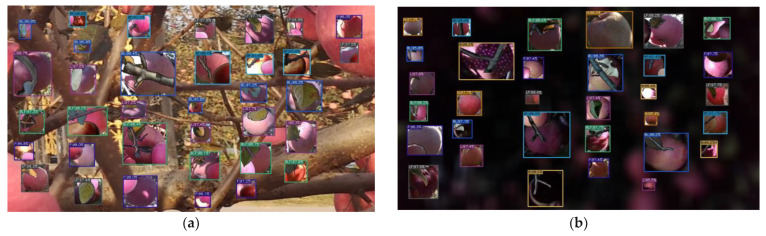
Synthesis image in high and low illuminations: (**a**) Synthesis image in high illumination; (**b**) Synthesis image in low illumination.

**Table 1 sensors-22-06325-t001:** Information about raw images of apple orchards.

Data	Owner	Country	Location	Acquisition Date	Sensor	Platform	Development Stage	Size	No. of Image
SD_ZY_IMG	CAAS	China	ZhaoYuan	25 October 2021	Intel d455	Handheld	filling-ripening	1280 × 720	4870
SD_QX_IMG	CAAS	China	Qixia	10 October 2020	Osmo Action1	Handheld	filling-ripening	640 × 480	2827
WT_PSR_IMG	Fu3lab	America	Prosser	2017/2018	Kinect V2	Tractor	filling-ripening	640 × 480	1558

**Table 2 sensors-22-06325-t002:** MTOA dataset statistics by high and low illuminations.

Data Name	No. of Images	No. of Annotation Boxes	No. of N	No. of B	No. of L	No. of BL	No. of BF	No. of F	No. of LF	No. of BLF
ZY_H	2738	36,803	9856	10,579	5597	6405	1936	1296	474	660
ZY_L	2132	25,165	9639	5671	4558	3133	728	901	252	283
QX_H	2681	8159	1273	3053	804	1442	934	268	116	269
QX_L	146	491	68	215	34	112	41	6	0	15
PSR_H	1558	10,774	2150	189	5295	942	54	540	1457	147
PSR_L	0	0	0	0	0	0	0	0	0	0

**Table 3 sensors-22-06325-t003:** Information for each model.

Model	Epochs	Image_Size	Original Augmentation Algorithm	New Augmentation Algorithm	Training Time (h)	Project Website
yolox-s	≤300	640 × 640	mosaic, mixup, flip, hsv	mosaic, mixup (≤60 epoch) flip, hsv (≥0 epoch)	7.8/17.2	https://github.com/MegEngine/YOLOX (accessed on 30 December 2021)
yolov5-s	≤300	640 × 640	mosaic, mixup, flip, copy_paste, multi-scale	flip, copy_paste, multi-scale	4.4/10.4	https://github.com/ultralytics/yolov5 (accessed on 30 November 2021)
yolov4-s	≤300	640 × 640	hsv, flip, mosaic	hsv, flip	11.1/27.7	https://github.com/WongKinYiu/PyTorch_YOLOv4 (accessed on 30 November 2021)
yolov3-tiny	≤300	640 × 640	flip, hsv, mosaic, mixup	flip, hsv	4.8/11.6	https://github.com/ultralytics/yolov3 (accessed on 30 November 2021)
efficientDet-d0	≤200	640 × 640	--	--	19.4/33.8	https://github.com/zylo117/Yet-Another-EfficientDet-Pytorch (accessed on 31 December 2021)

**Table 4 sensors-22-06325-t004:** Synthesis results of basic training dataset by high and low illumination.

Results of Data Synthesis	Total Synthetic	No. of N	No. of B	No. of L	No. of BL	No. of BF	No. of F	No. of LF	No. of BLF
ZY_H_syn	4,7829	723	0	4982	4174	8643	9283	10,105	9919
ZY_L_syn	59,467	940	4908	6021	7446	9851	9678	10,327	10,296
QX_H_syn	76,473	9306	7526	9775	9137	9645	10,311	10,463	10,310
QX_L_syn	0	0	0	0	0	0	0	0	0
PSR_H_syn	73,858	8429	10,390	5284	9637	10,525	10,039	9122	10,432
PSR_L_syn	0	0	0	0	0	0	0	0	0

ZY_H_syn, ZY_L_syn, QX_H_syn, QX_L_syn, PSR_H_syn, and PSR_L_syn represent the synthesized data after the equalization of ZY_H, ZY_L, QX_H, QX_L, PSR_H, and PSR_L, respectively.

**Table 5 sensors-22-06325-t005:** Detection results of different models on the test dataset.

Model Comparison	P	R	mAP_0.5_	AP_N_	AP_B_	AP_L_	AP_F_	AP_BL_	AP_BF_	AP_LF_	AP_BLF_
yolox-s-basic	0.894	0.845	0.892	0.904	0.898	0.897	0.892	0.886	0.888	0.884	0.885
yolox-s-bal	** *0.974* **	** *0.972* **	0.919	0.909	0.909	0.908	0.909	0.908	0.908	0.906	** *0.994* **
*diff*	0.080	0.127	0.027	0.005	0.011	0.011	0.017	0.022	0.020	0.022	0.109
yolov5-s-basic	0.947	0.914	** *0.960* **	0.984	0.979	0.97	0.960	0.957	0.948	0.955	0.934
yolov5-s-bal	0.946	0.936	** *0.960* **	** *0.985* **	** *0.982* **	** *0.975* **	** *0.963* **	** *0.965* **	** *0.959* **	** *0.956* **	0.898
*diff*	−0.001	0.022	0	0.001	0.003	0.005	0.003	0.008	0.011	0.001	−0.036
yolov4-s-basic	0.900	0.751	0.727	0.835	0.797	0.768	0.699	0.742	0.693	0.639	0.646
yolov4-s-bal	0.961	0.913	0.907	0.941	0.930	0.916	0.910	0.904	0.920	0.879	0.859
*diff*	0.061	0.162	0.180	0.106	0.133	0.148	0.211	0.162	0.227	0.240	0.213
yolov3-tiny-basic	0.856	0.791	0.883	0.959	0.936	0.941	0.856	0.889	0.842	0.877	0.768
yolov3-tiny-bal	0.910	0.842	0.918	0.97	0.95	0.956	0.897	0.912	0.895	0.917	0.835
*diff*	0.054	0.051	0.035	0.011	0.014	0.015	0.041	0.023	0.053	0.040	0.067
efficientdet-d0-basic	0.933	0.896	0.930	0.951	0.938	0.936	0.923	0.923	0.934	0.919	0.913
efficientdet-d0-bal	0.940	0.923	0.935	0.951	0.948	0.939	0.934	0.935	0.935	0.934	0.901
*diff*	0.007	0.027	0.005	0.000	0.01	0.003	0.011	0.012	0.001	0.015	−0.012

The *diff* represents the difference between detection metrics of the model trained by the balanced MTOA dataset and the model trained by basic training dataset, and the value is shown in bold black.

**Table 6 sensors-22-06325-t006:** Comparison of the statistical results of yolov5-s-basic and yolov5-s-bal for all metrics in test dataset.

Model Comparison	Data	P	R	mAP_0.5_	AP_N_	AP_B_	AP_L_	AP_F_	AP_BL_	AP_BF_	AP_LF_	AP_BLF_
yolov5-s-basic	ZY_H_TEST	0.943	0.935	0.960	0.983	0.975	0.964	0.944	0.971	0.940	0.977	0.926
yolov5-s-bal		0.970	0.950	0.970	0.986	0.983	0.970	0.968	0.979	0.957	0.970	0.944
*diff*		**0.027**	**0.015**	**0.010**	**0.003**	**0.008**	**0.006**	**0.024**	**0.008**	**0.017**	**−0.007**	**0.018**
yolov5-s-basic	ZY_L_TEST	0.954	0.913	0.96	0.985	0.966	0.950	0.968	0.935	0.952	0.953	0.971
yolov5-s-bal		0.966	0.943	0.965	0.992	0.983	0.979	0.977	0.951	0.985	0.967	0.886
*diff*		**0.012**	**0.030**	**0.005**	**0.007**	**0.017**	**0.029**	**0.009**	**0.016**	**0.033**	**0.014**	**−0.085**
yolov5-s-basic	QX_H_TEST	0.948	0.950	0.958	0.985	0.985	0.955	0.935	0.977	0.962	0.967	0.890
yolov5-s-bal		0.943	0.902	0.973	0.992	0.986	0.982	0.954	0.981	0.964	0.967	0.961
*diff*		**−0.005**	**−0.048**	**0.015**	**0.007**	**0.001**	**0.027**	**0.019**	**0.004**	**0.002**	**0.000**	**0.071**
yolov5-s-basic	PSR_H_TEST	0.923	0.936	0.960	0.989	0.985	0.983	0.964	0.947	0.905	0.963	0.942
yolov5-s-bal		0.957	0.886	0.934	0.979	0.951	0.982	0.964	0.936	0.859	0.957	0.846
*diff*		**0.034**	**−0.050**	**−0.026**	**−0.010**	**−0.034**	**−0.001**	**0.000**	**−0.011**	**−0.046**	**−0.006**	**−0.096**

The *diff* represents the difference between detection metrics of yolov5-s-basic assay and detection metrics of yolov5-s-bal, and it is bolded in black; the boxed area shows a larger decline in AP_BLF_.

**Table 7 sensors-22-06325-t007:** Comparison of the statistical results of yolov4-s-basic and yolov4-s-bal for all metrics in the test dataset for each subcategory.

Model Comparison	Data	P	R	mAP_0.5_	AP_N_	AP_B_	AP_L_	AP_F_	AP_BL_	AP_BF_	AP_LF_	AP_BLF_
yolov4-s-basic	ZY_H_ TEST	0.894	0.845	0.892	0.904	0.898	0.897	0.892	0.886	0.888	0.884	0.885
yolov4-s-bal		0.974	0.972	0.919	0.909	0.909	0.908	0.909	0.908	0.908	0.906	0.994
*diff*		**0.08**	**0.127**	**0.027**	**0.005**	**0.011**	**0.011**	**0.017**	**0.022**	**0.02**	**0.022**	**0.109**
yolov4-s-basic	ZY_L_ TEST	0.947	0.914	0.960	0.984	0.979	0.97	0.96	0.957	0.948	0.955	0.934
yolov4-s-bal		0.946	0.936	0.960	0.985	0.982	0.975	0.963	0.965	0.959	0.956	0.898
*diff*		**−0.001**	**0.022**	**0.000**	**0.001**	**0.003**	**0.005**	**0.003**	**0.008**	**0.011**	**0.001**	**−0.036**
yolov4-s-basic	QX_H_ TEST	0.900	0.751	0.727	0.835	0.797	0.768	0.699	0.742	0.693	0.639	0.646
yolov4-s-bal		0.961	0.913	0.907	0.941	0.93	0.916	0.910	0.904	0.92	0.879	0.859
*diff*		**0.061**	**0.162**	**0.180**	**0.106**	**0.133**	**0.148**	**0.211**	**0.162**	**0.227**	**0.240**	**0.213**
yolov4-s-basic	PSR_H_TEST	0.856	0.791	0.883	0.959	0.936	0.941	0.856	0.889	0.842	0.877	0.768
yolov4-s-bal		0.910	0.842	0.918	0.97	0.950	0.956	0.897	0.912	0.895	0.917	0.835
*diff*		**0.054**	**0.051**	**0.035**	**0.011**	**0.014**	**0.015**	**0.041**	**0.023**	**0.053**	**0.040**	**0.067**

The *diff* represents the difference between detection metrics of yolov4-s-basic assay and detection metrics of yolov4-s-bal, and it is bolded in black; the boxed area shows a larger decline in AP_BLF_.

## Data Availability

MTOA dataset can be accessed from the following link: https://github.com/oceam/MTOA (accessed on 1 July 2022).

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
