# Peer review of "Augmentation Method for High Intra-Class Variation Data in Apple Detection"

_sensors, 2022, doi:10.3390/s22176325_

Round 1

Author Response

Dear reviewer.

Thank you for your opinions, these comments are very helpful to improve the quality of the manuscript. We have elaborated on the way the MTOA dataset is used, presented the highlights of the paper, carefully added training time in table 4, and added to the Limitation and Future work section. In addition, we responded the reviewers' comments with a point by point and changes in revised manuscript. Full details of the files are listed. We sincerely hope that you find our responses and modifications satisfactory and that the manuscript is now acceptable for publication.

Comment 1:

 I could not find the apple dataset of multi-type occlusion of apple fruits which the authors mentioned in line 97. The link of dataset is empty. https://github.com/oceam/MTOA.

Response 1:

Thanks for your comment. we are working on the automatic push function of dataset download link on https://github.com/oceam/MTOA|, the reader only needs to fill out a very simple form to get the link of the dataset.

For reviewer check, here is the link:

https://www.dropbox.com/sh/zeb65hy4xlijtyf/AABPh2DVi_ExQjCHOB2I6Im-a?dl=0

Comment 2:

The novelty of this paper is not clear. The difference between present work and previous Works should be highlighted. What is novelty of this paper? Only the data balance augmentation algorithm seems, as mentioned in line 100.

Response 2:

The novelty can be summarized as comparing with data augmentation algorithms that use superposition, scaling, rotation, mixup and mosaic method among datasets, the proposed augmentation method is focused on multi-type occlusion datasets. The highlights of this section are described in lines 392-407.

All highlights are explicitly stated in lines 96 to 104 of the article, the highlights of this study are as follows:

  1. We created the MTOA, the first dataset considering multi-type occlusion of apple fruits, and made it available for free under the MIT license.
  2. We proposed a balance augmentation method to achieve a balanced number of annotation boxes of each apple occlusion class in different regions under different illuminations and solved the problem of severe differences in the number of samples between classes;
  3. We validated the effectiveness of the proposed algorithm using five popular lightweight object detection models. This is a new direction of data augmentation for the training of various lightweight models in the future.

Comment 3:

 The data balance augmentation algorithm is well explained in Section 2.2.1.

Response 3:

Thank you for your valuable suggestions. Section 2.2.2, 2.2.3, 2.2.4, 2.2.5, and 2.3 are the details of overall framework, which is based on the logic of original data production, data equalization method, equalized data This part is written according to the logic of original data production, data synthesis method, new dataset generation after synthesis method, and then model training and validation, where data balance augmentation algorithm is composed of 2.2.3, 2.2.4, and 2.2.5.

Comment 4:

Supplementary materials written in lines 179, 333, 443, and 463 are unavailable.

Response 4:

Thanks for your comment. These additions are in the Supplementary Material (sensors-1845275-supplementary), which contains Figure S1, Figure S2, Figure S3, Method 1 and each related notes.

Comment 5:

Please add training time for each model in Table 4.

Response 5:

We appreciate it very much for this good suggestion, and we have done it according to your ideas. We made changes in Table 4, mainly adding a column for model training time. The model training time is measured in hours and each model contains two training times, the first value is the training time using the MTOA dataset and the second value is the training time using the balanced MTOA dataset. We have also added the corresponding notes in lines 357-361 of the paper.

Comment 6:

Why did you choose 640*640 image size for training models?.

Response 6:

Thanks for your comment. Since all network models uniformly support 640*640 image input, we resize the images to 640*640 when we input them to the model.

Comment 7:

A more detailed evaluation and comparison of state of the art would be appropriate. The author only quantitatively evaluates, while qualitative evaluation is also essential. I suggest adding the results of the visual experiment.

Response 7:

Thanks for your comment. we have conducted visual image processing experiments and qualitative evaluation, all of which are in the supplementary materials, please refer to Figure S1, Figure S2, Figure S3.

Comment 8:

What is the limitation of the proposed model?

Response 8:

The limitation of this algorithm is that the model training time increases with the growth of the amount of data after augmentation, which is due to the fact that the dataset is greatly increased after augmentation, resulting in an increase in the overall training time. We have added this part to the discussion section in lines 490-494.

Comment 9:

Add a new section named “Limitation and Future work” to give a limitation of the proposed method and future research gaps in this field.

Response 9:

Thanks for reviewer’s comments, we have revised in “Limitation and Future work” part of the manuscript as “Despite the aforementioned achievements[40], there is still room for improvement in the proposed method. Firstly, the process of component pool production consumes much time because all elements need to be manually selected or extracted, such as selecting base images, manually segmenting fruits, leaves, branches, etc., which increases the cost of data production. Secondly, after the MOTA dataset is balanced, the size of the balanced MOTA dataset will increase, and the training time of the model will increase accordingly when more data is used for model training. In the future, our research direction is to reduce the dataset production time and labor cost by using the existing segmentation model for building component pool. This“Limitation and Future work” part can be seen in lines 496-504 of the article.

Comment 10:

I recommend to review the following papers

  1. Automatic Fire Detection and Notification System Based on Improved YOLOv4 for the Blind and Visually Impaired.

Response 10:

I received your valuable suggestions and read the paper, then I added  “Limitation and Future work” part for my paper on the way as it described and finally added the relevant parts to our paper and we cite this paper at last.

Reviewer 2 Report

In this study, the authors address the problem that most lightweight models detect multiple types of occlusion targets inefficiently during fruit picking, by proposing a multi-type occlusion apple dataset and a balance augmentation method.

The topic is very interesting and also the obtained results.

The authors describe very clearly the problem, the related works and the proposed method.

Both the dataset (which can be freely accessed via the link provided by the authors) and the experimental results are well described and validate the proposed approach. 

Moreover, a comparison with the state-of-the art object detection networks has been conducted to demonstrate the improvement achieved with the proposed method.

For there reasons, I think that this work is suitable for publication.

Minor comments:

- At page 2,lines 56-58, the authors state: "Therefore, fine detection of multi-type occlusion fruits ... [27]". I think the citation [27] is interesting but not consistent with the sentence.

- At page 2, lines 73-75, the authors state:"To reduce the overfitting problem ... SMOTE [37] methods". I think that the citation for SMOTE is wrong. Maybe, the right reference is:

Chawla, Nitesh V., et al. "SMOTE: synthetic minority over-sampling technique." Journal of artificial intelligence research 16 (2002): 321-357.

Moreover, please explain the acronym SMOTE before using it.

- In Tables 7 and 8 the boxed area shows the larger decline in AP_BLF. According to this notation, I think it is not clear for the reader what kind of results the boxed area in Table 6 shows. Please fix the notation in the tables or better explain Table 6.

- Repeated reference in [29] and [30]. Please remove one.

Author Response

Dear reviewer.

Thank you for the editor and reviewers’ opinions, these comments are very helpful to improve the quality of the manuscript. We have carefully corrected mainly errors in citations and references, and ambiguities in table 6. Now we response the reviewers' comments with a point by point and changes in revised manuscript. Full details of the files are listed. We sincerely hope that you find our responses and modifications satisfactory and that the manuscript is now acceptable for publication.

Comment 1:

At page 2, lines 56-58, the authors state: "Therefore, fine detection of multi-type occlusion fruits ... [27]". I think the citation [27] is interesting but not consistent with the sentence.

Response 1:

Thanks for reviewer’s comments, there was indeed a reference error here, and the reference [27] has been removed.

Comment 2:

At page 2, lines 73-75, the authors state:"To reduce the overfitting problem ... SMOTE [37] methods". I think that the citation for SMOTE is wrong. Maybe, the right reference is:

Chawla, Nitesh V., et al. "SMOTE: synthetic minority over-sampling technique." Journal of artificial intelligence research 16 (2002): 321-357.

Moreover, please explain the acronym SMOTE before using it.

Response 2:

Thanks for reviewer’s comments, a citation error occurred here, and our incorrect citation was removed, as detailed in lines 56-58, and adjustments were made to the references.

Comment 3:

 In Tables 7 and 8 the boxed area shows the larger decline in AP_BLF. According to this notation, I think it is not clear for the reader what kind of results the boxed area in Table 6 shows. Please fix the notation in the tables or better explain Table 6.

Response 3:

Thanks for reviewer’s comments, the ambiguity arises here, and I have adjusted the original expression in Table 6 accordingly, adjusting the font of the emphasis to bold italic to represent the highest values of each metrics in the detection results.

Comment 4:

Repeated reference in [29] and [30]. Please remove one.

Response 4:

Thanks for reviewer’s comments, we are very sorry for our careless mistake, and we have removed the duplicate references, as detailed in lines 599-600[28].

Round 2

Reviewer 1 Report

The authors responded fully to the comments and developed the manuscript. I think that the manuscript can be accepted in the present form.